# Feasibility Testing of the Health4LIFE Weight Loss Intervention for Primary School Educators Living with Overweight/Obesity Employed at Public Schools in Low-Income Settings in Cape Town and South Africa: A Mixed Methods Study [note 1]

**DOI:** 10.3390/nu16183062

**Published:** 2024-09-11

**Authors:** Fatima Hoosen, Mieke Faber, Johanna H. Nel, Nelia P. Steyn, Marjanne Senekal

**Affiliations:** 1Health through Physical Activity, Lifestyle and Sport Research Centre (HPALS), FIMS International Collaborating Centre of Sports Medicine, Division of Physiological Sciences, Department of Human Biology, Faculty of Health Sciences, University of Cape Town, Cape Town 7935, South Africa; hsnfat006@myuct.ac.za; 2Non-Communicable Diseases Research Unit, South African Medical Research Council, Cape Town 7505, South Africa; mieke.faber@mrc.ac.za; 3Centre of Excellence for Nutrition (CEN), North-West University, Potchefstroom 2530, South Africa; 4Department of Logistics, Faculty of Economic and Management Sciences, Stellenbosch University, Stellenbosch 7602, South Africa; jhnel@sun.ac.za; 5Department of Human Biology, Faculty of Health Sciences, University of Cape Town, Cape Town 7935, South Africa; marjanne.senekal@uct.ac.za

**Keywords:** healthy eating, behaviour change, belief patterns, teachers, weight management, self-help

## Abstract

Given the high prevalence of overweight and obesity amongst educators, this study investigated the feasibility of the 16-week Health4LIFE weight loss intervention for primary school educators living with overweight/obesity in low-income settings in Cape Town, South Africa. The research comprised two sub-studies, a pilot randomised controlled trial testing the intervention (10 intervention, *n* = 79 and 10 control schools, *n* = 58), and an investigation of the perceptions of participating educators and principals. Feasibility outcomes included reach, applicability, acceptability, implementation integrity, and a hypothesis-generating signal of effect on lifestyle factors and weight. The intervention consisted of a wellness day, weight loss manual, and text messages. Results indicated acceptable reach, with positive feedback on intervention components from principals and educators. Implementation was largely successful, though three schools dropped out due to scheduling issues. Barriers included interruption of teaching time and busy school schedules. The intervention group (*n* = 42) showed favourable shifts in belief patterns, stages of change, and lifestyle behaviours, with a trend towards weight loss. Control group (*n* = 43) changes were limited to dietary intake. The triangulation of results supported the intervention’s feasibility in terms of primary and secondary outcomes. Recommendations for enhancement include adding in-person follow-up sessions and an app-based element to potentially increase impact on lifestyle indicators and weight loss.

## 1. Introduction

Many African countries are facing widespread increases in obesity rates linked to the rapid demographic, socio-cultural and economic transitions occurring in these countries [1]. The same scenario is evident in South Africa, where the prevalence of overweight and obesity (BMI ≥ 25) was found to be 31.3% in men and 67.6% in women across a national survey conducted in 2016 [2]. Overweight and obesity are major risk factors for non-communicable diseases (NCDs) such as cardiovascular diseases, diabetes, musculoskeletal disorders and some cancers, and are the leading cause of mortality worldwide [1].

Surveys conducted amongst educators employed at public schools in low socio-economic areas in the Western Cape and other areas in South Africa have found that they may be at a higher NCD risk than the general South African population, with overweight and obesity rates as high as 85% being reported [3,4,5]. Despite the high prevalence of overweight and obesity among educators and the inherent risk it holds for NCDs, there is a paucity of research on health promotion interventions for educators not only in South Africa but also globally. In their systematic review on the effectiveness of health promotion interventions targeting obesity prevention in school-based staff, Hill et al. (2022) concluded that there are extensive gaps in evidence globally and emphasised the need for further research across all contexts to gain insights into effective interventions in this influential workforce [6].

The emphasis on health interventions for educators is not only for their own personal benefit, but also for the benefit of children they teach. Numerous studies have found that educators do not only promote learner motivation in aspects of education, but also life in general, and that they are not simply providers of education but role models of healthy behaviours, a positive attitude, and providers of support to learners [7,8,9]. It is therefore important that educators are aware of and held responsible for the health messages they impart, whether these messages are actively passed on as part of the curriculum, or in a more passive manner as part of their personal health behaviour, such as having a healthy weight, healthy eating pattern and being physically active.

Weight loss interventions should focus on achieving a modest weight reduction of 5% to 10% within 6 months, which has been shown to be associated with reduced blood pressure, reduced blood cholesterol and improved glycemic control in individuals living with overweight or obesity, even if they remain overweight or obese after weight loss [10,11,12,13]. Although this benefit is further increased with a greater amount of weight loss over an extended period, modest weight loss is readily achievable and should be the initial goal for obesity management [14]. There is consensus that the cornerstone of treatment for individuals living with overweight or obesity is a comprehensive lifestyle approach. This approach integrates a healthier dietary intake, physical activity components, as well as measures to support behavioural change [10,11,12,13,15].

We followed a theory and evidence-based approach, as recommended by the MRC guidelines [16,17], to develop a sustainable weight loss intervention that meets the specific needs of educators within the South African context, Health4LIFE. The development process was guided by a comprehensive and coherent framework, namely the Behaviour Change Wheel (BCW), which is underpinned by a behaviour change model (the Capability, Opportunity, Motivation—Behaviour Model) [18]. The BCW was further expanded upon and integrated with the Theory of Planned Behaviour (TPB) [19], the Health Belief Model (HBM) [20] and the use of the Step approach to Message Design and Testing (SatMDT) [21], strengthening the theoretical framework and making this intervention development approach unique. In addition, use of the TPB provided a deeper understanding of the educators’ beliefs regarding their dietary intake and physical activity, which directly informed the development of the Health4LIFE intervention in a novel and tailored approach. The aim of the present study was to investigate the feasibility of the Health4LIFE weight loss intervention for primary school educators living with overweight or obesity employed at public schools in low-income settings in the Western Cape, South Africa, using a mixed methods approach.

## 2. Materials and Methods

This paper conforms to the CONSORT extension guideline for randomised pilot and feasibility trials [16].


**Identification and definition of feasibility indicators**


According to Eldridge et al. (2016), feasibility can be viewed “as an overarching concept, with all studies done in preparation for a main study open to being called feasibility studies, with pilot studies being a subset of feasibility studies” [22]. Feasibility outcomes (elements) adopted for the present study were aligned with the approach used by Duijzer et al. (2014) and are summarised in Table 1 [23].


**Feasibility testing design**


The feasibility of the Health4Life weight loss intervention was investigated in a mixed methods design, combining a qualitative and quantitative approach in two sub-studies. Sub-study 1 investigated the reach and implementation integrity (primary outcomes) and signal of effect (secondary outcome) of the intervention, while Sub-study 2 was qualitative using semi-structured in-depth interviews to investigate primary intervention feasibility outcomes. Sub-study 2 investigated the perceptions of educators who participated in the intervention arm, as well as the perceptions of principals regarding reach, acceptability, applicability and implementation integrity (Figure 1). This feasibility trial was registered in the Pan African Clinical Trials Registry (PACTR) (trial number: PACTR201910535635886).


**Sub-study 1**


A.
Study design, target population, sample size selection and recruitment


The study design for this feasibility sub-study was a clustered pilot randomised controlled trial (RCT). Inclusion criteria were educators with a BMI ≥ 27 kg/m^2^, employed at select public schools within the Metro North District of Cape Town, English-literate, between 20 and 60 years of age, owning their own personal mobile phone and willing to be allocated to either the intervention or control group. Exclusion criteria were educators currently on a weight loss diet, women who were pregnant or trying to conceive during the study period, women who had given birth within 6 months prior to data collection, individuals with a history of major medical problems such as heart disease (including having had a pacemaker inserted) or those with recent weight loss greater than 10% in the previous six months.

Sample size estimation for feasibility testing is not based on power calculations, as is the case for the full-scale effectiveness testing of an intervention in a RCT, but rather on the specific objectives of the study [22]. There is agreement that if there is no hypothesis testing, a smaller sample size than that required for a full effectiveness study is appropriate [24,25,26,27]. The sample size calculation for a full effectiveness RCT to test the impact of the Health4LIFE intervention resulted in an estimate of 108 participants per arm (216 in total) to achieve 90% power at a 5% significance level, based on expected differences in weight change from previous weight loss trials [28,29,30,31,32,33,34]. For this feasibility study, we aimed to recruit 120 educators, 60 per arm, which is 55% of the calculated size for a full effectiveness trial.

Public schools within the selected district were stratified according to socio-economic status based on respective quintiles. Public schools across South Africa are grouped into five quintiles according to the demographics of the neighbourhood, where schools in quintile one are the poorest and those in quintile five are the least poor [35]. A school was regarded as eligible if it fell within quintiles three to five and if it had more than 20 educators employed at the school.

The random sampling of schools was repeated until the target of 20 schools was achieved. A total of 683 (150 male and 533 female) educators were employed across the 20 schools that were included in this study. Ten schools were then randomly assigned to the control and 10 to the intervention group. Schools and educators were blinded to the randomisation. The strategy used to recruit educators involved the hosting of wellness days, at intervention and control schools.

B.
Intervention: Health4LIFE weight loss intervention


*Intervention development:* An evidence-based and sustainable Health4LIFE weight loss intervention that meets the specific needs of educators within the South African context was developed using a comprehensive and systematic approach underpinned by behaviour theory, as was outlined in the introduction. The intervention was developed by Dr F Hoosen, with a panel of esteemed expert researchers providing oversight and guidance. Panel members included Professors M. Senekal; N. Steyn and A. de Villiers (senior researchers and experts in behaviour theory); Professor M. Faber (chief specialist scientist); Professor D. Evans (Professor of Prevention and Community Health and Global Health) and Mrs S. Booley (dietitian with master’s level research experience).

The outcome was a self-help Health4LIFE weight loss intervention that consisted of three elements, namely an initial face-to-face contact session as part of a wellness day conducted at schools to create health awareness, a hard copy self-help manual to facilitate dietary pattern and physical activity behaviour change and text messages sent over a 16-week period.

*Element 1 of the intervention (wellness day):* The wellness day was the first point of contact with educators, for awareness creation according to the HBM (first element in the intervention), to increase motivation, influence decision-making behaviour and self-efficacy [20]. The wellness day was managed by a registered dietitian and aimed to achieve the following: (1) creating health awareness by performing a NCD risk assessment, which included measuring weight and height and calculating body mass index (BMI), measuring blood pressure and waist circumference and determining percentage body fat, and (2) inspiring and motivating educators to take action and participate in the intervention based on the risk assessment findings.

*Elements 2 and 3 of the intervention:* Educators in the intervention group received a hard copy manual along with a set of 80 text messages, five per week for 16 weeks. The manual focused on general weight management and healthy eating advice, as well as specific dietary advice regarding increasing fruit and vegetable intake and decreasing fat and sugar intake, and increasing physical activity and stress management. Self-directed activities were also included in the manual. A detailed outline of the manual and text messages is provided in the Appendix A. The extent to which participants engaged with the self-help manual during the intervention period was not monitored.

C.
Control


The control group received the same NCD risk assessment as the intervention group as well as a hard copy of a National Department of Health (DoH) 10-page booklet that provides brief detail on a healthy lifestyle. These participants received a hard copy of the intervention manual at the end of the study period.

D.
Measures



*Survey questionnaire*


A self-administered questionnaire was developed for the purposes of this research to assess the following variables:

Socio-demographic data: Information obtained included date of birth, gender, age, years teaching and ethnicity. In addition, living standards were also investigated using the Living Standards Measure (LSM), which is a widely used marketing research tool in Southern Africa [36]. The LSM categorises the population into 10 LSM groups, where LSM 1–4 is categorised as ‘least access to wealth’ while LSM 8–10 is categorised as ‘most access to wealth’.

Weight-loss readiness to change: The stage of change, according to the transtheoretical model [37], was determined for five target behaviours, i.e., increase fruit intake, increase vegetable intake, decrease fat intake, decrease sugar intake and increase physical activity. For these purposes, educators had to select one of five statements that best described their readiness to change, i.e., pre-contemplative: no plans to change for the next 6 months, contemplative: thinking about changing in the next 6 months, preparation: thinking about changing in the next month, action: attempting change currently, and maintenance: changed behaviour and attempting to maintain change. The five stages of change as outlined in the stages of change framework was used to guide the development of these statements by an esteemed expert panel (MS, MF, NS, FH).

Level of physical activity*:* The Global Physical Activity Questionnaire (GPAQ) was used to assess physical activity (work, transport, leisure activity and time spent in sedentary behaviour) and comprises 16 questions [38]. It is commonly used to classify the level of physical activity as high, moderate or low and quantifies it in terms of Metabolic Equivalents (METs).

Beliefs relating to key dietary and physical activity behaviours: Educators’ beliefs relating to the consumption of fruit and vegetables (five control and one behavioural belief), sugar (three control and one behavioural belief) and fat (four control and three behavioural beliefs), as well as physical activity (six control and one behavioural belief), were investigated at baseline and 16-week follow-up (refer to Table 2 for the beliefs). Educators indicated whether they strongly disagreed, disagreed, are neutral, agreed or strongly agreed with each belief statement on a 5-point Likert scale. The belief statements were derived from formative work that involved a survey amongst 164 educators from schools in lower socio-economic areas in the Cape Metropole to identify salient beliefs to be targeted in the Health4LIFE intervention [39].


*Anthropometric assessments*


Weight: Body weight was measured with a digital scale to the nearest 0.1 kg. The scale (Seca, 813 Electronic Flat Scale, Hamburg, Germany) was placed on an even, uncarpeted area. Educators were weighed barefoot, wearing light clothing.

Height: Height measurements were carried out barefoot and read to the nearest 0.5 cm from a fixed standard stadiometer (Seca Leicester Height Measure, Hamburg, Germany) placed on an even, uncarpeted area. Educators faced the front, looking straight ahead along the Frankfort plane, with shoulder blades, buttocks and heels touching the measuring board.

Weight and height measures were repeated and the average of readings recorded.

Body mass index (BMI): Educators’ BMI was calculated as weight/height^2^ (kg/m^2^) and classified according to WHO criteria as either pre-obese (25–29.9 kg/m^2^), obese class I (30.0–34.9 kg/m^2^), obese class II (35.0–34.9 kg/m^2^) or obese class III (≥40.0 kg/m^2^) [40].


*Dietary intake assessment*


An existing food item list that was used to assess the food choices of educators in the Western Cape with a similar socio-economic status background [41] was adapted to meet the needs of the present study, resulting in a short 59-item, non-quantified indicator food list to measure food choices at baseline and 16-week follow-up. Educators were requested to sort a set of photocards of each item on the food list into a pile of those consumed and those not consumed in the previous seven days. The frequency of consumed items over the past seven days was subsequently determined. Food items were categorised into eight indicator food categories: high-fat foods, low-fat foods, sugary foods, refined carbohydrates, fibre-rich foods, fruit, vegetables and alcohol. The total frequency of intake per day from each indicator food category was determined by first converting the frequencies recorded as weekly to daily frequency (divided by 7) and, second, summing all frequencies per day within each category. The assigned foods of each indicator food category is as follow: **Low-fat food items**: Low-fat red meat, chicken without skin, low-fat milk/sour milk/yoghurt, fat-free/low-fat cheese/cheese spreads and low-fat salad dressing/mayonnaise; **High-fat food items**: High-fat red meat, processed meats, tinned meats, chicken with skin, organ meats, full cream milk/sour milk/yoghurt, full-fat cheese, peanut butter/peanuts, full-fat salad dressing/mayonnaise, pies/sausage rolls/samosas, crisps, crackers, take-outs, roasted or fried potatoes or chips, other fried food, margarine/butter on bread, margarine/butter in porridge and margarine/butter/oil added to vegetables; **Sugary food items**: chocolates, sweets, cakes/biscuits/doughnuts, juice nectar, other juice, canned fruit or stewed fruit, fizzy drinks, energy drinks, milky drinks, jam/marmalade/chutney, sugar/honey in tea/coffee, sugar/honey on cereal/porridge, and sugar/syrup added to vegetables; **Fibre-rich carbohydrates**: legumes and brown/wholewheat bread/rolls; **Refined carbohydrates:** white bread/rolls; **Fruit items:** Oranges/naartjies, any other fruit items such as apples/pears/bananas, dried fruit; **Vegetable items**: Orange/yellow vegetables, green vegetables, mixed vegetables, cabbage/cauliflower/lettuce/cucumber, salad and tomato; **ETOH items**: Alcoholic coolers and spirits/brandy/vodka with mixer.

E.
Data collection procedures


Data were collected at baseline and follow-up, 16 weeks after baseline. Fieldworkers were trained and methods were standardised to obtain the necessary anthropometric and clinical measures, conduct the food choice interview and facilitate the completion of the self-administered questionnaires. Fieldworkers were not blinded to the participants’ group allocation.


**Sub-study 2**


A.
Study design


The study design for the investigation of the perception of educators and principals of the intervention was qualitative, using goal-directed, semi-structured in-depth interviews which were conducted face-to-face. In-depth interviews can provide context to other data (such as outcome data) by offering a more complete account of events during an intervention and exploring the reasons behind it, thereby creating a deeper understanding of the data [42,43].

B.
Study samples


The target populations first included educators who had completed the 16-week Health4LIFE weight loss intervention across the eight schools, subcategorised based on sex. The principle of data saturation was applied to determine the final number of interviews [44,45], which was achieved after six interviews had been conducted. Secondly, school principals from the participating intervention and control schools that provided permission for the research were targeted. Data saturation in the case of principal interviews was achieved after four interviews had been conducted [44,45].

C.
Interview guides


The focus of the interview guide for educators was on their experience of the intervention as a whole, and then more specifically of the manual and the text messages. The interview guide for the principals aimed to elicit information on how they experienced the implementation process of the intervention and the recruitment strategy used, as well as their perceptions of the intervention itself and how it was experienced by the educators. Principals were also afforded an opportunity to suggest how the research approach could be improved. The interview guides were pilot tested on one educator and one principal. Data from these two interviews were included in the final data analysis, as no changes needed to be made to the two intervention guides (Appendix A).

D.
Data collection procedures


In-depth interviews were conducted by a single interviewer (FH), who was trained and whose methods were standardised for these purposes by MS, who has extensive qualitative research experience. Interviews were recorded with an audio digital recorder. At the end of the interview, the interviewer provided the educator with a synopsis of the discussion to ensure that everything that was recorded was correct and to provide them with the opportunity to add any further perspectives. These interviews took approximately 45 min. The data collection period commenced after the 16-week intervention period ended.


**Data management and analysis**


Quantitative data: Quantitative data were obtained for one primary feasibility outcome, namely reach, and for all secondary outcomes relating to potential impact such as change in weight, including BMI; food choices; physical activity; sedentary behaviour; and beliefs relating to fruits and vegetables, sugar and fat intakes and physical activity. Data were entered into a Microsoft Excel (2007) spreadsheet and analysed using Statistica (Version 13.5.0.17). Frequencies were tallied for categorical data. The mean with standard deviation (SD) or median with interquartile range (IQR) were calculated for numerical data depending on normality of the data. As the belief scores included count variables, the median (IQR) was calculated and used in all analyses involving beliefs.

The change in secondary outcome variables were tested by protocol. Within-group (control group and intervention group) change from baseline to 16-week follow-up for these variables was tested using the paired t-test for dependent samples or Wilcoxon matched pairs test, depending on the normality of the data. The comparison of within-group change over the 16-week intervention period between the control and intervention groups was carried out using the independent samples t-test or Mann- Whitney U test depending on the normality of the data.

A principal component analysis (PCA) (SAS Version 9.4, SAS for Windows; SAS Institute, Cary, NC, USA) was conducted to generate belief patterns regarding dietary intake (fruit and vegetable intake, sugar intake, and fat intake) and physical activity for study completers. The components were extracted using a principal axis method which was followed by a varimax rotation. The factors considered in the determination of the number of belief patterns to be retained included the visual observation of the scree plot, an eigenvalue >1 for the number of factors to retain, the factor pattern loadings, and the interpretability of patterns as performed by Sprake et al. (2018) to identify dietary patterns of university students in the United Kingdom [46]. Three factors (patterns) were extracted from the intervention and control group data for each time point. Beliefs with a pattern loading of >0.40 on a belief pattern were retained to interpret the pattern. Of note is that a universally accepted cut-off for factor loading does not exist. Tables that list each belief on the three belief patterns, the factor loading of each belief, and present Kaiser’s measure of sampling adequacy and percentage variance, explained by each pattern for the intervention and control groups at baseline and follow-up, are presented in Appendix A

Results with a *p*-value of <0.05 were deemed to be statistically significant and are presented in bold font in Table 3.

Qualitative data: The data were transcribed by the first author. A deductive thematic analysis was then carried out by hand. An iterative approach based on the six steps as outlined by Braun and Clark [47] was followed: (1) familiarisation with the data, (2) coding, (3) identifying themes, (4) reviewing themes, (5) refining and naming themes and (6) presenting results. Transcripts were coded and initially analysed by the first author (FH). To ensure reliability, a second researcher (MS) conducted quality control by reviewing all transcripts and their associated codes and themes. Where discrepancies occurred, the researchers discussed the coded transcripts and themes until reaching a consensus.

## 3. Results

### 3.1. Sub-Study 1

#### 3.1.1. Reach of the Intervention

Of the 683 (22%, *n* = 150 male and 78%, *n* = 533 female) educators employed across the 20 schools, 349 (51.2%) attended the wellness days, of whom 140 (40.1%) volunteered to participate in the study. After further screening, the final number of educators who enrolled in the study was 137 (intervention group *n* = 79; control group *n* = 58), 39.3% of those who attended the wellness day (Figure 2). The total drop-out rate was 38.0% (*n* = 52), 46.8% (*n* = 37) from the intervention group and 25.9% (*n* = 15) from the control group. Of the total drop-out group, 59.6% (*n* = 31) were from three schools (2 intervention schools, *n* = 26 and 1 control school, *n* = 5) that did not allow a follow-up visit to be arranged. In total, 85 educators completed the study, 42 from the intervention group and 43 from the control group. A flow diagram of the study sample is presented in Figure 2.

#### 3.1.2. Baseline Profile of Completers for the Intervention and Control Groups

The median (IQR) age of the intervention group was 48.0 (22.6–58.8) years and the control group was 45.6 (23.0–60.1) years. Most educators in both the intervention (92.9%) and control groups (88.4%) were female. The intervention group was significantly more likely than the control group to include black educators than white educators (intervention group: 40.5% black, 9.5% white; control group: 20.9% black, 32.6% white) (Pearson’s Chi Square *p* = 0.018). Within the intervention group, black educators were significantly more likely to drop out than coloured educators (Pearson’s Chi Square *p* = 0.042). Most educators in the intervention and the control group were reportedly married (59.5% and 55.8%, respectively), non-smokers (90.5% and 90.7%, respectively), and fell into the higher LSM category (8–10) (81% and 93%, respectively).

The median (IQR) number of teaching years was 17.0 (1.0–40.0) for the intervention group and 14.0 (0.6–37.0) for the control group (not significantly different; Mann–Whitney U test: *p* > 0.05). Within the intervention group, educators who had previously tried to lose weight were significantly more likely to drop-out than those who had not tried to lose weight before (Person’s Chi Square *p* = 0.039). The majority of participants in the intervention and control group were in the obese class I or II category (61.9% and 55.8%, respectively).

#### 3.1.3. Change from Baseline to 16-Week Follow-Up in Secondary Outcome Variables

There were no significant differences between the intervention and control groups for any of the secondary outcome variables at baseline (Table 3). There were also no significant differences between the two groups for within-group change from baseline to 16-week follow-up for any of the variables (Table 3). There was a trend for the intervention group to have lost weight (*p* = 0.093) and have a reduced BMI (*p* = 0.086), which was not evident in the control group (*p* = 0.299 and *p* = 0.32, respectively).

There was a significant increase in moderate recreational MET minutes per week from baseline to 16-week follow-up within the intervention group (Table 3). There was no significant within group change in time spent being sedentary in either the intervention or control group; at 16-week follow-up, the intervention group spent significantly less time being sedentary than the control group (Table 3).

There were significant increases in the frequency of intake of low-fat food items as well as vegetables within both the intervention and control groups. There were no significant within-group changes for the frequency of intake of high-fat food items, fruit, fibre-rich carbohydrates and refined carbohydrates. There was a significant decrease in the frequency of consumption of sugary food items, the frequency of adding fat to foods and the frequency of adding sugar to foods within the intervention group, but not within the control group.
nutrients-16-03062-t003_Table 3Table 3Change in anthropometry, physical activity level and dietary intake in the intervention and control groups from baseline to 16-week follow-up.Indicator Baseline16-Week Follow-UpFollow-Up—Baseline ^1^ControlInterventionControlInterventionControlInterventionAnthropometryMedian (IQR)*n* = 43Median (IQR)*n* = 42Median (IQR)*n* = 43Median (IQR)*n* = 42Median (IQR)*n* = 43Median (IQR)*n* = 42Height (m)1.64 (1.48–1.95)1.60 (1.43–1.74)**N/A**Weight (kg)86.8 (72.5–148.7)93.8 (64.7–146.4)87.5 (72.6–147.0)93.5 (64.0–145.4)−0.50 (−13.35–6.00)*p* = 0.299 ^2^−0.51 (−3.82–3.78)*p* = 0.093 ^2^BMI kg/m^2^32.2 (27.1–54.3)34.9 (27.6–54.1)32.4 (26.9–54.6)34.9 (27.6–54.3)−0.19 (−4.50–2.63)*p* = 0.320 ^2^−0.21 (−1.70–1.49)*p* = 0.086 ^2^**Physical activity ^3^*****n*****(%)*****n*****(%)*****n*****(%)*****n*****(%)****N/A**MET minutes ≥600/week16 (48.5)11 (30.6)15 (45.5)13 (36.1)
**Median (IQR)*****n* = 33****Median (IQR)*****n* = 36****Median (IQR)*****n* = 33****Median (IQR)*****n* = 36****Median (IQR)*****n* = 33****Median (IQR)*****n* = 36**Total MET min/week300.0 (0.0–11,280.0)0.0 (0.0–8400)240.0 (0.0–19,200.0)340.0 (0.0–18,840.0)0.0 (−8400.0–8400.0)0.0 (−8400.0–18,840.0)Moderate recreational MET min/week0.0 (0.0–1920.0)**0.0 (0.0–240.0)**0.0 (0.0–1440.0)**0.0 (0.0–2000.0)**0.0 (−600.0–840.0)**0.0 (0.0–2000.0)*****p* = 0.003 ^2^****Within group**Sedentary time min/day240.0 (60.0–900.0)180.0 (20.0–840.0)**300.0 (0.0–720.0)****180.0 (10.0–480.0)**0.0 (−540.0–330.0)0.0 (−360.0–360.0)***p* = 0.043 ^4^****Between group****Food groups****Median (IQR)*****n* = 43****Median (IQR)*****n* = 42****Median (IQR)*****n* = 43****Median (IQR)*****n* = 42****Median (IQR)*****n* = 43****Median (IQR)*****n* = 42**Low fat food items/day**0.8 (0.0–3.9)****0.9 (0.0–5.2)****1.2 (0.0–15.0)****1.3 (0.2–6.8)****0.6 (−1.5–12.8)*****p* < 0.01 ^2^****Within group****0.4 (−1.5–5.0)*****p* = 0.002 ^2^****Within group**Sugary food items/day1.7 (0.0–4.9)**1.9 (0.0–8.7)**1.5 (0.0–4.0)**1.6 (0.2–4.3)**−0.2 (−3.0–2.6)**−0.2 (−7.4–1.0)*****p* = 0.010 ^2^****Within group**Vegetable items/day**1.6 (0.0–4.7)****1.4 (0.0–3.3)****1.8 (0.2–8.0)****1.9 (0.4–10.5)****0.4 (−2.9–4.6)*****p* = 0.043 ^2^****Within group****0.7 (−1.4–7.8)*****p* = 0.002 ^2^****Within group**Fat added to food/day1.0 (0.0–4.0)**1.5 (0.2–7.0)**0.7 (0.0–4.7)**0.8 (0.0–14.0)**−0.2 (−2.8–3.4)**−0.5 (−5.6–12.3)*****p* = 0.011 ^2^****Within group**Sugar added to food/day1.0 (0.0–4.2)**1.5 (0.0–6.2)**0.4 (0.0–8.0)**0.8 (0.0–8.0)**−0.2 (−4.2–6.0)**−0.3 (−3.4–8.0)*****p* < 0.01 ^2^****Within group**IQ: Interquartile range; m: metres; kg/m^2^: kilograms per square metre; MET: Metabolic Equivalent; min: minutes; IQR: Interquartile range; N/A: not applicable. Significantly different results are presented in bold font. ^1^ Difference between intervention and control group for within-group change: Mann–Whitney U test. However, there were no significant differences for any listed variable for within-group change between the intervention and control group. ^2^ Within group change from baseline to follow-up: Wilcoxon matched pairs test. ^3^ N varies due to missing values. ^4^ Between-group difference at follow-up: Mann–Whitney U-test. **Low-fat food items**: Low-fat red meat, chicken without skin, low-fat milk/sour milk/yoghurt, fat-free/low-fat cheese/cheese spreads and low-fat salad dressing/mayonnaise; **High-fat food items**: High-fat red meat, processed meats, tinned meats, chicken with skin, organ meats, full cream milk/sour milk/yoghurt, full-fat cheese, peanut butter/peanuts, full-fat salad dressing/mayonnaise, pies/sausage rolls/samosas, crisps, crackers, take-outs, roasted or fried potatoes or chips, other fried food, margarine/butter on bread, margarine/butter in porridge and margarine/butter/oil added to vegetables; **Sugary food items**: chocolates, sweets, cakes/biscuits/doughnuts, juice nectar, other juice, canned fruit or stewed fruit, fizzy drinks, energy drinks, milky drinks, jam/marmalade/chutney, sugar/honey in tea/coffee, sugar/honey on cereal/porridge, and sugar/syrup added to vegetables; **Fibre-rich carbohydrates**: legumes and brown/wholewheat bread/rolls; **Refined carbohydrates:** white bread/rolls; **Fruit items:** Oranges/naartjies, any other fruit items such as apples/pears/bananas, dried fruit; **Vegetable items**: Orange/yellow vegetables, green vegetables, mixed vegetables, cabbage/cauliflower/lettuce/cucumber, salad and tomato; **ETOH items**: Alcoholic coolers and spirits/brandy/vodka with mixer.


Table 4 presents a summary of the three belief patterns and key focus areas identified for the intervention and control groups at baseline and 16-week follow-up. Table 2 shows the numbered, linked beliefs. In the intervention group, a notable shift in the first two patterns from baseline to follow-up is evident, where the focus areas within each belief pattern became more focused and linked to the information provided in the intervention. At baseline, the strongest (first) pattern was predominantly focused on beliefs related to physical activity and the health benefits of a healthy lifestyle, without any focus on healthy food choices in patterns one or three and only a limited focus on fat intake in pattern two. At follow-up, a shift in the belief patterns was evident for the intervention group. The strongest pattern now reflected a more comprehensive representation of a healthy lifestyle, focusing on facilitators of healthy eating and physical activity, as well as the capacity to implement these behaviours. The second pattern focused on the health benefits resulting from a healthy lifestyle, such as better weight control. However, the intervention group maintained four barrier beliefs from baseline to follow-up, with three loading onto pattern three: Belief 12: I do not have enough time to prepare healthy meals regularly; Belief 18: there are no accessible, safe, affordable opportunities for me to be physically active; Belief 16: reducing the amount of sugary foods/snacks/drinks I eat and drink will make me feel unwell (moody or have a headache or tired) and Belief 9: low-fat/healthy fat options are expensive.

In contrast, the control group did not exhibit a similar shift in belief patterns at follow-up. The control group’s strongest (first) pattern at baseline involved the health benefits of and behaviours related to healthy food choices. This shifted to a strong focus on health benefits of a healthy diet and physical activity, with less focus on the actual behaviours and facilitators thereof. Patterns 2 and 3 both emphasised the health behaviours related to a healthy lifestyle and facilitators thereof at baseline, which shifted to a focus on physical activity (pattern 2) and healthy food choices (pattern 3). The control group had no barrier beliefs at baseline, but two barrier beliefs emerged at follow-up: Belief 9: low fat/healthy fat options are expensive (pattern 1) and Belief 12: I do not have enough time to prepare healthy meals regularly (negative loading on pattern 2). (Table 3 and Table 4).

The percentage of educators in the intervention group who were either “attempting change currently” or had “changed behaviour and attempting to maintain change” for the behaviour “*increase fruit intake*” increased from 52% at baseline to 80.9% at 16-week follow-up. For the control group, this was similar at baseline and 16-week follow-up (44.5% and 48.9%, respectively). At 16-week follow-up, the intervention group was significantly more likely to be “attempting change currently” than the control group for this behaviour (Pearson’s Chi Square Test: *p* = 0.028) (Figure 3a).

The percentage of educators in the intervention group who were either “attempting change currently” or had “changed behaviour and attempting to maintain change” for the behaviour “*decrease sugar intake*” increased from 69% at baseline to 76.1% at 16-week follow-up. For the control group, there was an increase from 51.2% at baseline to 70.2% at 16-week follow-up. At 16-week follow-up, the intervention group was significantly more likely to be “attempting change currently” than the control group for this behaviour (Pearson’s Chi Square Test: *p* = 0.031) (Figure 3d).

As far as the stage of change for vegetable and fat intake (Figure 3b,c) and physical activity (Figure 4) is concerned, it is evident that there were no significant within groups differences between time points, with more than 60% in both intervention and control groups contemplating change in the next month or currently attempting to change at both time points.

### 3.2. Sub-Study 2

Six in-depth interviews were conducted with female educators who completed the Health4LIFE intervention. Nine themes emerged, namely ‘Wellness Day’, ‘School environment’, ‘Barriers’, ‘Facilitators’, ‘Goals’, “Manual critique’, ‘Text message critique’, Contact frequency’ and ‘Delivery’. Four interviews with school principals were conducted, with two principals from intervention schools that completed the intervention, one from a control school that completed the study and one from a control school that dropped out of the study. Six themes emerged, namely ‘Educator wellness’, ‘Wellness day’, ‘Barriers’, ‘Facilitators’, ‘School-based weight loss interventions’ and ‘Contact Frequency’. Quotes supporting the themes for both the educators and the principals are presented in Table 5.

#### 3.2.1. Wellness Day

According to the educators, the wellness day evoked feelings of wellbeing, excitement and even feeling pampered in some. It also created personal health awareness, with some describing it as a wake-up call. Principals’ perceptions were that educators were mostly positive about the wellness day, that they felt pampered, that it was beneficial, that they were appreciative of it and that it created health awareness.

The principals themselves also found that the wellness day created health awareness and they regarded the strategy of using the wellness day to implement the intervention in a positive light.

#### 3.2.2. Weight Loss Intervention in the School Setting

It was apparent that educators saw great value in having a weight loss intervention implemented at their place of work. They expressed positive feelings about such an intervention, specifically citing the value of peer support and the convenience and ease of accessing an intervention at the workplace. Workload emerged as a big contributor to educators’ stress levels.

The principals mostly viewed a school-based weight loss intervention in a positive light, and that they thought it was convenient and contributed to peer support and may assist educators with overcoming any barriers to engaging with the intervention successfully.

#### 3.2.3. Barriers

The educators reported several factors as barriers to a self-help weight loss intervention, of which time seemed to be one of the strongest barriers. Specific time-related aspects included time to follow the intervention in general, not having enough time for themselves and not having time to follow a meal plan. Finances (budget) also emerged as a barrier to following the intervention in general, as well as following the meal plans. The educators also mentioned that completing the activities in the manual was challenging.

Factors mentioned by principals as barriers to a weight loss intervention included the interruption of teaching time, educators perceiving the intervention as another task for them to complete within the context of already having a full school programme, while the after-school scheduling of the intervention would also be problematic as it could coincide with personal responsibilities or prior commitments that cannot be rescheduled; a further barrier that emerged was the fact that schools have to apply to the Department of Education if they want to deviate from the normal school day.

#### 3.2.4. Facilitators

From the interviews with educators, support from family and peers emerged as a strong facilitator for following the weight loss intervention. The text messages were indicated to be motivational, helpful and serve as a reminder to follow the intervention. It was further mentioned by some that they felt that the intervention recommendations involved only minor changes to the diet which could facilitate compliance.

Principals identified two key factors that could facilitate intervention implementation and mitigate loss of teaching time: carefully planned visits and class supervision by non-teaching staff.

#### 3.2.5. Health Goals of Educators

Educators mentioned fat loss, weight loss and a healthy diet as being their goals immediately after the wellness day. Goals which emerged at the time of the interview include a healthy lifestyle and weight loss, but also not wanting to lose weight. Other goals which were also mentioned include following a healthy diet, feeling healthy and feeling more energetic.

#### 3.2.6. Principals’ Perceptions on Educator Wellness

A strong theme that emerged was that principals thought that environmental factors play a huge role in the wellness of educators. The environment was mentioned in general, but more specific factors were also mentioned and included economic and social conditions, a heavy workload, under-performing students and stress. Principals also considered educator wellness to be important and that educator wellness was associated with improved outcomes (in general, for the educators, learners, and the school). Principals mentioned that educator wellness may be improved if educators had knowledge of a nutritious dietary intake and if a school considered educator wellness to be part of its curriculum. Principals perceived that if educator wellness was not addressed, it could result in negative implications, such as absenteeism, poor curriculum delivery, eating a non-nutritious diet and educators wanting to resign from their positions.

#### 3.2.7. Educators’ Critique of Manual and Text Messages

As far as critique of the hard copy manual to facilitate dietary pattern and physical activity for health and weight loss is concerned, it emerged that some educators found it useful (specifically the sections relating to tips and the meal plans), easy to understand and enjoyable (specifically the sections regarding tips, meal plans and the self-assessment exercises). Suggestions to improve the manual included considering providing it in other languages, reducing the size of the manual and adding more information to the eating plan and physical activity sections. While the text messages were sent in the afternoon, most of the educators indicated that mornings would be better.

#### 3.2.8. Frequency of Contact

It emerged that most educators felt that the frequency of contact that formed part of the self-help weight loss intervention (one contact session in the form of the wellness day) was inadequate. Their suggestions relating to frequency of contact that emerged included that quarterly visits should be included, that such visits could serve for checking health indicators and providing relevant feedback, and that such visits could potentially motivate educators to follow the intervention.

It clearly emerged that principals felt that frequency of contact in the Health4LIFE intervention was not sufficient and that regular quarterly health visits were needed. Principals mentioned that no contact during the 16-week intervention was not good as educators lost their focus in this time; it was also mentioned that educators needed more assistance during the intervention period.

#### 3.2.9. Delivery of Online Intervention

It appeared that educators viewed the possibility of an online delivery format of a weight loss intervention positively, specifically the younger generation.

### 3.3. Summary of Feasibility Outcome Measures

Table 6 presents a summary of the results of the feasibility assessment of the Health4LIFE intervention to inform recommendations for a future full-scale study.

## 4. Discussion

This study set out to investigate the feasibility of a self-help weight loss intervention that was developed for primary school educators living with overweight or obesity, employed at public schools in low-income settings in Cape Town. Results reflecting reach, acceptability, applicability and implementation integrity (primary or process outcomes), and hypothesis-generating signals of effect relating to weight loss and lifestyle indicators (physical activity, food choices, associated beliefs and stage of change) support the feasibility of the Health4LIFE intervention for full-scale testing or roll-out in the real-world setting.

Educators in this research were mostly female, in their mid-forties, in a higher living standard category (8–10), non-smokers and had a BMI above 30 kg/m^2^. Two-thirds of the baseline control group and just more than a third of the intervention baseline group had attempted weight loss before. Of note is that study drop-outs were significantly more likely to have attempted weight loss before than study completers.


*Primary outcomes*


Predominantly, the triangulation of results shows that the reach and applicability of the intervention support feasibility thereof, as reflected by the attendance of the wellness day and subsequent recruitment of educators, as well as the principals’ and educators’ feedback on the implementation of the weight loss intervention. However, the reach was hindered during the initial recruitment phase by the reluctance of principals to provide consent for the intervention. In a much earlier study (1998), where an educator wellness programme was implemented at elementary schools, a similar trend was found, where 82 primary school principals were invited to participate, and 22 principals immediately indicated that they were not interested [48]. Principals in the present study indicated that barriers to participation included the possibility that teaching time may be interrupted, prior school/personal commitments that could not be rescheduled, an already full school programme, and that schools require permission from the Department of Basic Education (DoBE) for any deviation from the normal school day.

In addition to these points mentioned by principals, educator-specific challenges within the South African school setting may reduce their willingness to sign up for an intervention, thus reducing reach and applicability. These include work dissatisfaction, personal health issues, exposure to violence and job-related stress [49,50]. In the present study, educators mentioned that an intervention, even if targeted at themselves, could be considered to be another task to add to their already full schedules. This is corroborated by findings from a 2004 survey among educators across South Africa that work overload was a factor within the school environment that impacted negatively on educator health and wellbeing [50]. Zuma et al. (2016) further reported that one of the main reasons cited by educators for wanting to leave the profession was a heavy workload [49]. However, there are also factors within the school setting which may support an uptake of interventions. These include available facilities (e.g., sports fields), and regular opportunities to engage in healthy behaviours (e.g., actively participating in sports lessons or activities aimed at learners), which may promote the long-term implementation of educator-targeted interventions at schools [51].

Workplace wellness days have previously been reported to be popular [52,53] and well-attended [52]. This strategy was used to create health awareness along the lines of the HBM [20] in the present study. Educators and principals expressed very positive feelings about the wellness day and more than half of educators in the 20 recruited schools opted to complete the NCD risk assessment. This supports the notion that the wellness day is an effective ”entry-into-care” approach in the school setting, as was also found by Joseph et al. (2018) [54].

The reach in terms of male educators participating in the study was in line with the percentage of male educators employed across the schools. It should, however, be noted that poor male recruitment into weight loss interventions seems to be a global phenomenon and may therefore require special attention. A recent systematic review that investigated the efficacy of workplace interventions to improve lifestyle behaviours of educators, reported that all the included studies (six) had higher proportions of females than males, up to 98% [55]. Franz et al. (2007) also reported that less than a fifth of weight loss participants are typically male [56]. A recent review of weight management interventions targeted at males classified as overweight or obese by Kim and Shin (2020) showed that males were less likely to partake in weight loss interventions as they appeared less concerned about their weight or less likely to attempt weight loss than females. They further considered weight loss interventions to be a part of the “female world” [57]. Another point of consideration is that the wellness day, which was the entry point of the intervention, was run by a female dietitian. In Sub-Saharan Africa, it has been shown that males are more likely to be recruited into interventions when the health service provider is a male [58]. It is thus imperative that intervention delivery mode, materials and the content of a weight loss intervention speak to and are attractive to male educators. Furthermore, the team involved in wellness days and potential follow-up should include male fieldworkers and/or intervention facilitators.

The reasonable retention rate achieved in the present study, despite educators lost to follow-up because researchers could not obtain permission from three of the study schools to conduct follow up assessments, is not uncommon for interventions occurring in the workplace, including schools [54]. A recent systematic review that investigated the efficacy of workplace intervention in the school setting reported a retention rate ranging from 30.3% to 100% [55]. Reasons for the higher retention rates in the workplace that have been mentioned include participants being actively followed-up if a session was missed [59], the close proximity of sessions being held at the worksite [59], the development of a “trusted clinician relationship” [59,60] and a low employee turnover rate, which makes it possible to incorporate regular follow-up visits in the intervention [61]. Educators in the present study expressed that they valued having the intervention within the school setting, specifically mentioning factors such as peer support, convenience and ease of access, which may have contributed to the retention rate.

It was evident that black African educators, who were mostly female, were more likely to drop out from the intervention group in the present study. Research has found that black African women in South Africa positively associate obesity with health, wealth and beauty [62,63,64]. It is therefore important that these community and cultural perceptions are considered when planning and implementing a weight loss intervention. Another factor that seems to have affected retention rates is previous weight loss attempts, which was significantly more likely in the intervention drop-out sub-group than the intervention completers group. A systematic review identified the most consistent pre-treatment predictor of weight loss to be fewer previous attempts at dieting [65]. These researchers suggested that previous weight loss attempts should be assessed before educators commence an intervention to identify those at risk for poor outcomes to ensure that they obtain assistance with increasing self-confidence and addressing barriers [65].

Acceptability of the intervention from the principals’ point of view seems to have been linked to their concern for educator wellness, and the prevention of consequences of not addressing health issues, including absenteeism, poor curriculum delivery, and eating a non-nutritious diet, and resignations. Their concerns about educator well-being are corroborated by the findings of a 2004 survey among educators across South Africa, which identified work dissatisfaction, work overload, personal health issues and exposure to violence as factors within the school environment that impact educator health and wellbeing negatively [50]. The negative health implications of stressful working conditions are further evident from a study conducted in Cape Town among educators who opted for permanent retirement due to psychiatric disorders. Most retirees in the study cited work-related stress as a major contributing factor to their illness [66].

Educators themselves felt that they would like to lead a healthy lifestyle, and some specifically wanted to focus on weight loss as a goal, thus also indicating acceptance of the intervention. Although educators mostly expressed positive critique about the manual content, some mentioned that completing the activities linked to each of the chapters was a challenge. These activities included, amongst others, the self-monitoring of dietary intake and physical activity, which has been shown to be associated with greater weight loss in a recent review of predictors of weight loss [67]. Non-compliance with self-monitoring activities by educators in self-help weight loss interventions is not uncommon [68]. Lee et al. (2018) explained that the successful completion of intervention activities reflects appropriate participant engagement and, consequently, intervention integrity [69]. Finding innovative solutions to overcome this challenge is therefore imperative. This may involve incorporating eHealth into the intervention, such as an electronic self-monitoring feature, which is reportedly more commonly used and quicker than the paper and pencil diary [70].

Educators were very positive about the text message element and indicated that it facilitated engagement with the intervention. Similarly, the value of such messaging has been demonstrated in other studies where text-message interventions resulted in good engagement with the intervention, leading to behaviour change [71,72]. However, tailoring the timing of messages to participants’ personal preferences should be considered, as shown in a recent smoking cessation study: if timing was judged to be optimal by participants, the messages provided support and motivation, but if judged to be sub-optimal, it caused irritation [73]. The integrity of the delivery of text messages in the present study was monitored to ensure that messages had actually been sent, that they had indeed been received and that the correct message was received at the correct time point by each educator.

The developers of the Health4LIFE weight loss intervention considered a self-help approach, with only one point of in-person contact at the start of the 16-week intervention as most appropriate to comply with the current DoBE policy, which only allows educators to leave the classroom for official school business [74]. Although educators and principals were accepting of such an intervention, it was evident that neither party approved of only one point of in-person contact. Nearly all interviewees indicated that more regular contact during the intervention period was preferred and that educators lost focus and/or motivation during the 16-week intervention period because there was no further contact. Nathan et al. (2020) reviewed the efficacy of workplace interventions for educators and also found educators preferred regular contact [55]. Duijzer at al. (2014) also reported that participants in their study cited monitoring and repeated measurements during the intervention as important to them, as it contributed to their motivation [23]. Tang et al. (2014) concluded in their systematic review of self-help interventions that personal contact was an important component of the interventions reviewed that resulted in weight loss [75]. Of note is that the consideration of inclusion of more frequent contact points is only feasible if endorsed by the DoBE and if funding is made available for these purposes.

The potential acceptability of an online platform for weight loss intervention delivery was also explored and interviewees expressed positive feelings in this regard. Delivery via an online platform could address the needs expressed for more contact, follow-up assessments and relevant feedback. In a recent systematic review of 12 studies, smartphone app features such as self-monitoring, personalised goal setting, feedback and reward systems, counselling and social support were found to increase self-regulation, and potentially promoted healthy behaviours and subsequent weight loss [76]. The coronavirus pandemic has also changed the way the world views communication over online platforms, which has now become an acceptable means of communication [77]. Alternative modes of delivery such as eHealth interventions, which support the online delivery of an intervention, should therefore be considered when in-person contact is limited or not possible, as recommended by leading health authorities [10,11,13].


*Secondary (effect) outcomes*


Feasibility testing is not powered to determine the effectiveness of the intervention in terms of weight loss and other outcome variables, but rather provides insights in whether there is potential to achieve specified secondary outcomes (signal of effect) [16].

Weight loss results in the present study show that the Health4LIFE weight loss intervention may have the potential to result in weight loss in educators living with overweight or obesity, as a non-significant trend toward weight loss was observed in the intervention group but not in the control group. If weight loss is found in full-scale testing, it may be partially explained by the contribution the intervention made to creating health awareness as specified in the HBM [20] and changes along the TPB pathway (beliefs, intention to change and performing target behaviours) [19].

The analysis of the belief results in the present study followed a novel approach by focusing on shifts in belief patterns rather than changes in individual beliefs. We postulated that healthy lifestyle beliefs are interconnected, and considering them individually might not reflect this interaction. At baseline, the intervention group’s first two belief patterns focused primarily on physical activity and the health benefits of a healthy lifestyle, with limited emphasis on food choices. At follow-up, this shifted to a comprehensive focus on a healthy lifestyle (pattern 1: healthy eating and physical activity facilitators and the behaviours thereof; and pattern 2: health benefits resulting from a healthy lifestyle). This shift reflects the Health4LIFE intervention’s emphasis on the four dietary (increase fruit and vegetable intake, reduce fat and sugar intake) and two physical activity behaviours (increase physical activity, reduce sitting time), as well as the health benefits resulting from a healthy lifestyle. Patterns in the control group were not as well balanced at follow-up. Their strongest pattern focused on health benefits that result from healthy food choices, while beliefs about physical activity moved to the second pattern, and healthy food choices only appeared in the third (weakest) pattern. The 10-page DoH booklet they received was less comprehensive, did not target specific beliefs and proportionally covered physical activity guidance in much more depth than dietary intake.

Notably, while the intervention group maintained four barrier beliefs regarding healthy eating and physical activity from baseline to follow-up (with three loading onto the weakest pattern), the control group showed a different trend. At baseline, they had no barrier beliefs, but at follow-up, they developed two barrier beliefs regarding healthy food choices, loading onto patterns 1 and 2, respectively.

Intention to change, which is necessary to facilitate behaviour change [19], was not assessed directly in the present study. Instead, shifts in the stage of change for target behaviours were investigated as proxy indicators of intention. Support for this approach comes from a Norwegian study of a South Asian population, where the relationship between a lifestyle intervention and stages of change for healthy eating [78] and weight loss were investigated [79]. Upward movement through change stages was reflected in the changes made in dietary intake [78]. In addition, Kjøllesdal et al. (2011) showed that being in the action stages for several healthy dietary habits at follow-up was related to weight loss, regardless of the stage at baseline [79].

In the present study, the intervention group profile for the stage of change for the behaviour ‘*increase fruit intake*’ shifted from ‘*no plans to change*’ or ‘*thinking about changing*’ towards ‘*currently attempting to change*’ or ‘*having changed*’ and were significantly more likely to be in the ‘*action phase*’ at follow-up than the control group. The shift in the stage of change in the intervention group was, however, not accompanied by an increased frequency of consumption of fruit (there was also no change in the control group). For the behaviour ‘*increase vegetable intake*’, the shift in the stage of change was similar in the intervention and control groups, with profiles not being significantly different at follow-up. The shift in the stage of change was accompanied by a significant increase in the frequency of vegetable intake in both groups. It is possible that the shifts in the stage of change, especially for fruit intake in the intervention group, reflect the shift in belief patterns. Beliefs related to fruit and vegetable intake loaded on the strongest belief pattern in the intervention group, while the majority of fruit- and vegetable-related beliefs in the control group only loaded on the third (weakest) pattern. The combined fruit and vegetable intake of educators was less than the recommended five-a-day [80] at baseline and follow-up.

For ‘decrease fat intake’, the shift in the stage of change from ‘no plans to change’ or ‘thinking about changing in the next six months or next month’ towards ‘currently attempting to change’ or ‘having changed’ was similar in the intervention and control groups, with profiles not being significantly different at follow-up. The shift in the stage of change was accompanied by a significant increase in the frequency of intake of low-fat food items in both groups, as well as a significant decrease in frequency of adding fat to food in the intervention group only. It is possible that the shifts in the stage of change for fat intake especially in the intervention group, reflect the shift in belief patterns. Johansen et al. (2010) also found a shift in action stages in the intervention group from baseline to follow-up for the amount and type of fat to be consumed, which translated into an increased consumption of healthier types of fat [78]. Baseline results in both the intervention and control groups in the present study show that high fat foods were consumed three times more frequently than low-fat foods (less than once a day) and fat was added to food between once to twice a day, potentially reflecting a dietary pattern with poor food choices in terms of fat content.

For ‘decrease sugar intake’, the shift in the stage of change from ‘no plans to change’ or ‘thinking about changing in the next six months or next month’ towards ‘currently attempting to change’ or ‘having changed’ was less prominent in the intervention than control group. However, at follow-up, the intervention group was significantly more likely to be in the “action phase” than the control group, which was accompanied by a significant decrease in the intake of sugary food items and a significant decrease in frequency of adding sugar to food in the intervention group. This may be explained by the shift in belief patterns, where only the intervention group had a positive behavioural belief related to the ability to reduce sugar intake loading onto the strongest pattern. Johansen et al. (2010) also found an increase in the proportion of participants in the action stages observed from baseline to follow-up, only in the intervention group for sugar, which translated into a decreased sugar intake [78].

For ‘increase physical activity’, there was only a minimal shift in the stage of change from ‘no plans to change’ or ‘thinking about changing in the next six months or next month’ towards ‘currently attempting to change’ or ‘having changed’ in both the intervention and control groups, with the profiles not being significantly different at follow-up. This may explain the inadequate physical activity levels of the majority of educators in the intervention and control groups at baseline and follow-up. It is, however, important to note that the intervention group significantly increased their recreational activity from baseline to follow-up and spent significantly less time being sedentary than their control counterparts at follow-up. The structure and focus of the Health4LIFE intervention were based on research that has shown that although both healthy food choices and physical activity are the cornerstones of the treatment of overweight and obesity, a hypocaloric intake is the most important factor in weight loss interventions [81,82,83]. It is therefore important to ensure that the intervention does not create a perception that physical activity is more important than dietary behaviour, as illustrated in the control group where the physical activity beliefs loaded on the second belief pattern, while most of the healthy food choices beliefs only loaded onto the third (weakest) pattern.


**Strengths and limitations of the study**


The strengths of the present study included the fact that the Health4LIFE weight loss intervention was specifically developed and tested for the target population within the school setting, following the best intervention development practice guidelines as set by the UK MRC [23,24]. Furthermore, a novel approach was used in the analysis of the belief results to focus on shifts in belief patterns rather than changes in individual beliefs. A further strength is that primary outcome variable insights were gained from the triangulation of results from both quantitative and qualitative investigations.

Limitations include that the pilot study was not powered to test changes in beliefs, lifestyle behaviours or weight, but the results provide preliminary, hypothesis-generating insights into the potential of the intervention to bring about the required changes. Although the dietary intake questionnaire was not validated in the target population, the same instrument was used at baseline and at 16-week follow-up to investigate the signals of effect. The insights provided in the feasibility testing of the Health4Life weight loss intervention may not reflect the perceptions of male educators and drop-out schools, as none consented for sub-study 2. Self-reported data should always be interpreted with caution, as the information provided on socio-demographic data, health status, dietary intake and physical activity is prone to measurement error.

## 5. Conclusions and Recommendations

Results reflecting reach, acceptability, applicability and implementation integrity (primary outcomes) and signals of effect (secondary outcomes) of the Health4LIFE intervention support the feasibility of the intervention. The hypothesis-generating signals of effect included favourable shifts in belief patterns regarding healthy lifestyle behaviours and facilitators; favourable shifts in the stage of change for “increase fruit intake” and “decrease sugar intake”, significant changes in some lifestyle behaviours (increased intake in low fat food items, increased intake of vegetables, decreased intake of sugary food items, decreased frequency of adding fat food, decreased frequency of adding sugar to food, increase in physical activity and decreased time spent being sedentary) and a trend towards weight loss in the intervention group. The only significant changes in the control group related to dietary intake was an increased intake of vegetables and an increased intake of low-fat foods.

It is recommended that the following receive attention in terms of the refinement of the Health4LIFE intervention before it is either tested in a full-scale evaluation study or implemented in real world settings: the recruitment of male educators, the drop-out rate of black African educators and those who have attempted weight loss before, the lack of DoBE policies to address educator health and wellbeing, and educator suggestions to improve the intervention manual, including increased frequency of contact and poor completion of self-monitoring activities. It is further emphasised that any future school-based interventions must be endorsed by the DoBE to ensure that educator health interventions are accommodated as part of the school calendar and are included in the national schooling budget. The endorsement by principals is also important, as they play a role in encouraging educator participation in school initiatives, as was shown by Joseph et al. (2018) [54] and Pretorius and De Villiers (2009) [84]. Finally, it would be prudent to advocate for the inclusion of educators as a specific target group in the next revision of the South African Strategy for the Prevention and Management of Obesity.

## Figures and Tables

**Figure 1 nutrients-16-03062-f001:**
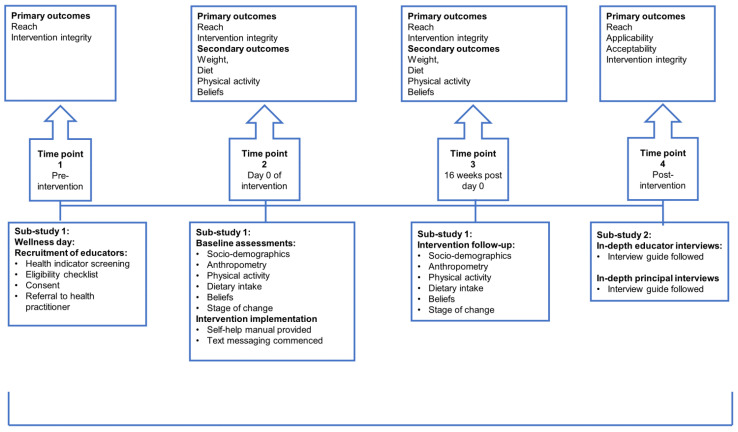
Summary of the feasibility testing design outlining assessments and associated primary and secondary outcomes of Sub-study 1 and 2.

**Figure 2 nutrients-16-03062-f002:**
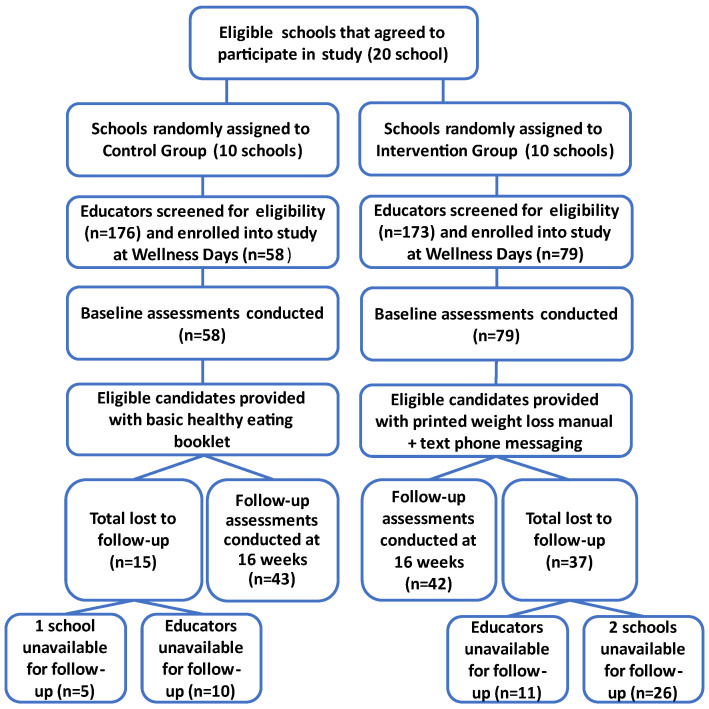
Randomisation of schools, recruitment of subjects and flow of assessments.

**Figure 3 nutrients-16-03062-f003:**
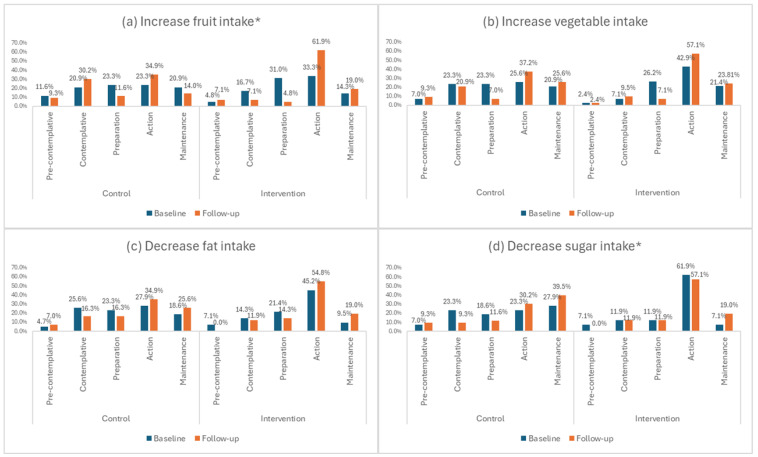
(**a**–**d**) Stage of change for dietary behaviours relating to fruit, vegetable, fat and sugar intake. * Significant difference between intervention and control groups at 16-week follow-up (Pearson’s Chi Square Test: *p* < 0.05).

**Figure 4 nutrients-16-03062-f004:**
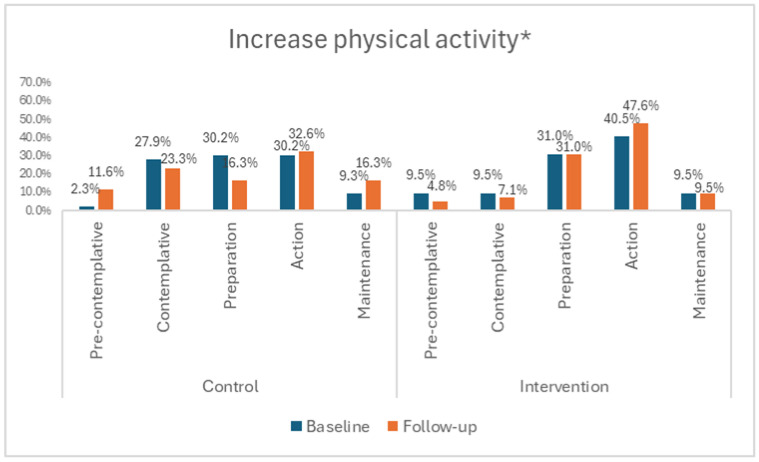
Stage of change relating to physical activity. * No significant differences between the intervention and control groups.

**Table 1 nutrients-16-03062-t001:** Definitions of feasibility outcomes for the present study.

Feasibility Elements	Definition for the Present Study
Reach (primary outcome)	Number of educators who (1) attended the wellness day, (2) were living with overweight or obesity and opted to participate in the intervention, and (3) completed the intervention.
Acceptability (primary outcome)	Degree to which educators and principals were happy with (1) a weight loss intervention within the school setting, (2) the wellness day at the school, (3) the content of the Health4LIFE intervention manual and text messages, (4) the mode of delivery and (5) the frequency of contact.
Applicability (primary outcome)	Degree to which the Health4LIFE intervention can be implemented within a school setting
Implementation integrity (primary outcome)	Was the intervention implemented as planned
Signal of effect (secondary outcome)	Potential impact of the intervention on target outcomes. Of note is that feasibility studies are not powered to test these outcomes [22].

**Table 2 nutrients-16-03062-t002:** List of salient beliefs (24) regarding fruit and vegetable intake, fat intake, sugar intake and physical activity targeted by the intervention.

Beliefs Regarding Fruit and Vegetable Intake (6 Beliefs)		
Preparation of vegetables does not take a long time.	2. * Eating fruits and vegetables every day will help me lose weight/control my weight.	3. Fruits and vegetables are affordable.
4. I can eat the recommended amounts of fruit and vegetables every day.	5. I would eat vegetables, even if at times they look unappealing.	6. Fruit and vegetables are easy to find in stores nearby.
**Beliefs Regarding Fat Intake (7 Beliefs)**		
7. * Eating less fat will help reduce the risk of diseases, e.g., heart disease.	8. * Decreasing the amount of fat I eat will help me lose/control my weight.	9. Low-fat/healthy fat options are expensive.
10. Healthy takeaways and/or street foods are easy to find in my surroundings.	11. It is easy to exclude high-fat foods from my daily diet.	12. I do not have enough time to prepare healthy meals regularly.
13. Low fat/fat-free foods taste good/are tasty.		
**Beliefs Regarding Sugar Intake (4 Beliefs)**		
14. I turn to sugary foods/snacks/drinks when I am stressed.	15. I have poor awareness of the sugar content in the food/drinks I eat/drink.	16. * Reducing the amount of sugary foods/snacks/drinks I eat and drink will make me feel unwell (moody or have a headache or tired).
17. I can reduce the amount of sugary foods/snacks/drinks I eat and drink.		
**Beliefs Regarding Physical Activity (7 Beliefs)**		
18. There are no accessible, safe, affordable opportunities for me to be physically active.	19. * Being physically more active will make me feel better about my appearance.	20. Finding time to be physically more active is possible.
21. Knowing more about different types of physical activity I can do will help me to be more active.	22. I could be more physical activity even if I were tired.	23. Having an exercise “buddy” will help me to be physically more active.
24. I can increase my levels of physical activity (be physically more active).		

Bold font and colour indicate beliefs regarding fruit and vegetable intake, fat intake, sugar intake and physical activity, respectively. * Health benefit or disadvantage of healthy lifestyle.

**Table 4 nutrients-16-03062-t004:** Summary of belief patterns and focus areas for the intervention and control groups from baseline to 16-week follow-up ^1^.

Intervention Group			
**Pattern**	**Baseline (KMO = 0.62)**	**Pattern**	**16-Week Follow-up (KMO = 0.70)**
	**Key Focus Areas Per Belief Pattern**		**Key Focus Areas Per Belief Pattern**
1 *	Physical activity: Behaviours and facilitators ** Healthy lifestyle: Health benefits	1 *	** Healthy lifestyle: Behaviours and facilitators
2	Healthy food choices: Behaviour and facilitator	2	** Healthy lifestyle: Health benefits
3	Healthy food choices: Barriers	3	** Healthy lifestyle: Barriers
**Control group**			
**Pattern**	**Baseline (KMO = 0.64)**	**Pattern**	**16-week follow-up (KMO = 0.63)**
	**Key focus areas per belief pattern**		**Key focus areas per belief pattern**
1 *	Healthy food choices: Health benefits and behaviours	1 *	Healthy food choices: Health benefits
2	** Healthy lifestyle: Behaviours and facilitators	2	Physical activity: Health benefit, behaviours and facilitators
3	** Healthy lifestyle: Behaviour and facilitator	3	Healthy food choices: Behaviours and facilitators

KMO = Kaiser’s measure of sampling adequacy. ^1^ Tables that list each belief on the three belief patterns, the factor loading of each belief, and present the Kaiser’s Measure of Sampling Adequacy and percentage variance explained by each pattern for the intervention and control groups at baseline and follow-up are presented in Appendix A. * First pattern is the strongest. ** Healthy lifestyle beliefs include a combination of beliefs related to food choices and physical activity.

**Table 5 nutrients-16-03062-t005:** Quotes illustrating the themes from the principal and educator interviews.

Wellness Day
**Educators** - “..*It set the mood for what is going to happen it made you excited and you actually looked forward to the program.”* (Educator 6) - “…*many were not aware of their health status and health readings…Screening was a big plus. Therefore many welcomed the program*.” (Educator 3)
**Principals** - “*But at the end they (teachers) feel very positive about it and it was a good intervention at the best time.*” (Principal 2)
**Weight loss intervention in the school setting (school environment)**
**Educators** - “*It makes things easier because now you are in a group and you not alone on your own because if you are on your own then you think Aghh you not going to do it. And it is convenient having it at school. If it is out of school, then really…I would not be able to do it*.” (Educator 4) - “*we are so stressed up with work at school and we are so loaded with lots of work that we need to do… So its difficult to set time aside (outside of work) to look after yourself*.” (Educator 1) **Principals**
- “ *I think it will be a good additive. And something that they know is available to them. Because you think in terms of moneywise, you must make an effort to get if you can get on your own. Here it is a normal, you must be here so you can as well partake in a program*.” (Principal 2) - “*The idea by having this program (at school), is I think is accepted by almost all staff at the school. So yes, I am very positive*.” (Principal 3)
**Barriers**
**Educators** - “*When we spoke about it more or less everybody was saying the same thing, they didn’t have time to follow the program*.” (Educator 1) - “*I really would like to eat more healthy and I learnt a lot of valuable lessons as to what I must eat but I don’t have the time for those things, I really don’t have the time to... the budget and the time. Those are the two things that made me stop*.” (Educator 2)
**Principals** - “*Anything which takes away from teaching time can be a barrier*.” (Principal 1) - “*So, during the course of the week every day is filled to capacity. So, when you came we had to cancel some of these things.*” (Principal 4)
**Facilitators**
**Educators** - “*Being part of a program with colleagues—we could share the experience as a team. I saw it as a sign to get more healthy. Team support was important to me.*” (Educator 3) - “*My husband’s support also helped a lot -he found an online exercise program for us and started*.” (Educator 3) - “*A (text) message just for the day to encourage you, to follow (the intervention).*” (Educator 1) - “…*there weren’t too many changes in your diet or daily eating plan*.” (Educator 5) - “*I monitored my weight and found it helpful and got excited when progress was made.*” (Educator 3) **Principals** - “*But then also let’s say you can come on a more regular basis then we can plan it (visits) much better... time management is therefore important so as not to interrupt teaching time too much*” (Principal 3) - “*We have a few students (Student teachers) at the school but we also have a few SGB (school governing body) appointed teachers and support staff that can actually at least be in the class while the teacher is gone*.” (Principal 3)
**Health goals**
**Educators** (after the wellness day) - “*Errm I realised that day (wellness day) that I was actually overweight and ya overweight for my height ya and ermm, yes, I wanted to lose fat because I learnt that being overweight has medical implications in the long run so ya*” (Educator 2) - “*I know I have horrible eating habits. So that was my reason. I wanted to improve my eating habits*.” (Educator 6)
**Principals’ perceptions on educator wellness**
- “*I think that the physical condition of teachers is of importance. For doing what they need to do, by producing their best for our kids, measuring it against the economic and also the social conditions that we are teaching in*.” (Principal 2) - “*And I think these social ills that we have, drugs, and FAS, alcohol syndrome kids, TB… So then I say a wellness program is for us, for our school, critically important*.” (Principal 2)
**Manual and text messages critique**
**Educators** - “*I found it very useful and easy to follow. I also found this (tips) useful and helpful—having the alternatives helped a lot.*” (Educator 3) - “..*everything that was at hand at the shops. Nothing was really difficult. I enjoyed it*”. (Educator 5)
**Frequency of contact**
**Educators** - “*It was too long. I think if you can come back perhaps every 2 weeks and check on us I think that would be great. Or once a month at least.*” (Educator 1) - “*It’s like you did this in the beginning and then we didn’t see you again. So maybe if you do it more regularly (health visits) because then we will say ok sister is coming tomorrow let’s just do our things*.” (Educator 2)
**Principals** - “Perhaps every 4 weeks someone can come and remind the group as to what the focus is” (Principal 3) - “*if you start a program you must assist the person to get into this thing. and develop it and become used to what he or she is doing to make it a lifestyle. Now that didn’t happen*.” (Principal 4)
**Delivery of online intervention**
**Educators** - “*I think it’s the younger generation now mostly they are on social media and whatever*” (Educator 1) - “*it also depends on who your clientele are because an older person may not like it*.” (Educator 2)

Colour and bold font indicates different themes while italics indicates direct quotes.

**Table 6 nutrients-16-03062-t006:** Summary of feasibility outcomes derived from sub-studies 1, 2 and 3 conducted as part of the feasibility testing of the Health4LIFE intervention.

Feasibility Outcome Measures	Indicators	Findings
Reach (quantitative and qualitative data) (Primary outcome)	Recruitment statistics	Two cycles of a random selection of 24 schools per cycle were needed to achieve the target of 20 participating schools because of the reluctance of principals to consent to the research.
	Principals’ opinions *	Principals mentioned the following potential barriers to participation: interruption of teaching time; prior commitments of educators which cannot be rescheduled; an already full school programme and the need to apply to the DoBE for deviations from the normal school day.
	Attendance of wellness day and recruitment statistics	The wellness day worked well to recruit the target number required for the purpose of this study.
	Gender representation	Only 20% of educators recruited into the study were male, but this is in line with the percentage of male educators employed across the schools.
	Retention statistics	The workplace-based intervention worked well to retain educators with a 38% total drop-out rate.
		Three schools did not allow for a follow-up visit as it was not convenient for them, and this accounted for 59% of the drop-out rate. Schools that allowed baseline and follow-up visits had good retention rates of 80%.
	Comparison of study completers with study drop-outs	African black educators were more likely to drop out. Educators who had attempted weight loss before were more likely to drop out (intervention group).
Acceptability (qualitative data) (Primary outcome)	Educator ** and principal * critique of a wellness day to recruit educators into the intervention	Principals and educators perceived the wellness day to be an acceptable strategy to recruit volunteers and create awareness of the intervention.
	Educator ** and principal * critique of a weight loss intervention within a school setting	Principals and educators perceived the school setting to be acceptable for implementation of a weight loss intervention for educators.
	Educator ** health goals (primary outcome)	Educators mentioned that a healthy lifestyle and weight loss was still a goal for them.
	Principal * critique of a weight loss intervention	Principals were accepting of implementing a weight loss intervention for educators, acknowledged the importance of educator wellness and mentioned that it was their perception that educators were also positive about the possibility of implementing a weight loss intervention in the school setting.
	Educator ** critique of the intervention content	Educators perceived the information provided in the manual and text messages to be acceptable. Recommendations to improve the manual included considering providing it in other languages, improving the structure of the manual and adding more information to the eating plan and physical activity sections. Educators also indicated they would like to select their preferred time to receive text messages.
	Educator ** critique of mode of intervention delivery	Educators perceived the intervention delivery (wellness day, hard copy manual to facilitate dietary pattern and physical activity for health and weight loss and text messages) to be acceptable.
	Educator ** and principal * critique of frequency of contact during the intervention period	Educators and principals felt that only one point of in-person contact during the 16-week intervention period was not acceptable.
Applicability (qualitative data) (Primary outcome)	Principal * critique of the implementation of the intervention within their school	Principals perceived that the implementation of a weight loss intervention within a school setting that has a full academic calendar could be challenging. Principals mentioned that support from the DoBE and flexibility when researchers arrange visits would facilitate implementing the intervention.
	Educator ** critique of the implementation of the intervention	Educators mentioned that an intervention could be perceived as another task for them to add to their already full schedules.
	Educator ** and principal * critique of frequency of contact during the intervention period (primary outcome)	Educators and principals mentioned that despite the challenges associated with an intervention within the school setting, they would prefer frequent health visits, even during the fourth school term.
Implementation integrity (quantitative and qualitative data) (Primary outcome)	Recruitment statistics at the wellness day	Eligible educators were identified at the wellness days, and 40.1% of those who attended agreed to participate in the intervention.
	Educator ** critique of engagement with intervention manual and text messages	Educators mentioned that they engaged with the manual and the text messages. Completing the self-monitoring activities was reported to be a challenge.
	Lost to follow-up (drop-out) statistics	The 16-week follow-up visit was not possible at three schools as a suitable time could not be arranged, resulting in 31 educators lost to follow-up.
	Measures implemented to ensure text messages were sent as intended	A service provider was sourced to send out the text messages over the 16 weeks. The PhD candidate was included on the recipient list to ensure that messages were received as intended.
Signal of effect (quantitative data) (Secondary outcome)	Changes in secondary outcomes (weight, diet, physical activity, beliefs, readiness to change,)	A trend towards weight loss within the intervention-, but not in the control group, was evident. Significant positive lifestyle changes within the intervention-, but not in the control group (decreased intake of sugary food items, decreased frequency of adding fat and sugar to food, increased physical activity and decreased sedentary time). Shifts in belief patterns regarding healthy lifestyle behaviours and facilitators that reflect the intervention content, but not in the control group. Significant positive changes in readiness to change within the intervention- but not control group for the behaviours “increase fruit intake” and “decrease sugar intake” (movement to attempting change currently or attempting to maintain change from contemplation of change).

DoBE: Department of Basic Education. * From in-depth interviews with four principals of participating schools. ** In-depth interviews with six educators who completed the intervention.

## Data Availability

The data presented in this study are available upon request to the corresponding author, pending ethical approval from the Faculty of Health Sciences Human Research Ethics Committee, University of Cape Town.

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
