# Peer review of "Feasibility Testing of the Health4LIFE Weight Loss Intervention for Primary School Educators Living with Overweight/Obesity Employed at Public Schools in Low-Income Settings in Cape Town and South Africa: A Mixed Methods Study†"

_nutrients, 2024, doi:10.3390/nu16183062_

Round 1

Reviewer 1 Report

Comments and Suggestions for Authors

This manuscript does an excellent job in all aspects of what is hoped for in a scholarly research article on an important topic of public health.

It is written clearly, provides very solid quantitative and qualitative findings, and addresses implications for policy and practice.

A small point is to ask if the authors would be willing to address how best to gain more gender-balanced participation in weight loss exercises.

Reviewer 2 Report

Comments and Suggestions for Authors

Dear Authors,

Thank you for your manuscript. Please see my comments below.

The Introduction section is rather brief. While the problem is presented well, the need for another intervention remains unclear. To date, many weight loss interventions have small and temporary effects. Thus, your Introduction requires a strong theoretical background for your weight loss intervention, explaining which behavioral change theories were employed to anticipate its effectiveness and advantage.

Sample size calculation in interventions is usually based on the estimated effect size. Please provide the calculation to support the information in lines 121-126.

In the study title, it is stated that this is a mixed-methods analysis, but in the Methods section, the type of mixed-methods study is not specified and detailed; only the quantitative and qualitative parts are described separately.

The description of the dietary intake assessment is unclear. The period of dietary recall (a day, a week) is not provided. The criteria for classification into food groups (high fat, low fat, etc.) are not specified. Thus, it is unclear what these groups represent.

Moreover, the purpose of conducting PCA and the significance of its results for this study (lines 299-310) are unclear. The description of the PCA is incomplete (data fit to PCA, the number of factors extracted, factor loadings, and the percentage of variance explained are not provided).

The reasons for the high percentage of dropouts are not explained. Also, the methods used to assess adherence to the intervention program are not provided.

Given these comments on study methods, the validity and interpretability of the results are questionable.

I suggest reducing the study methods and results for this paper, with a more precise description and analysis. This would make the paper more transparent and clear.

Finally, the paper demonstrates a high percentage of agreement—64% in total and 58% with one internet source. Copying and pasting from an internet source, even if you are the author of a PhD thesis, is considered unethical and is called self-plagiarism. Some parts should be reworked to avoid such a high percentage of agreement.

Comments on the Quality of English Language

Minor English issues were found.

Reviewer 3 Report

Comments and Suggestions for Authors

Housen et al present a work including 2 studies. No significant plagiarism has been detected and the language needs only minor editing.

The homogeneity of the populations, has not been addressed and
if yes not been presented.
Accuracy as well as power of the studies has to be calculated and added.

Comments on the Quality of English Language

language needs only minor editing

Reviewer 4 Report

Comments and Suggestions for Authors

Introduction

I believe that the approach of considering educators as models of healthy living should not justify the work. The focus should be on the worker's health. What are the social determinants of health for this group? What are the food environments in these schools?

Methods

I suggest following the CONSORT checklist recommendations in Substudy 1. Was a sample size calculation performed? I recommend calculating it based on the outcome and effect size. Was randomization conducted? Was there any type of blinding?

I suggest reviewing the use of the term “obese.” 

Was the method for assessing food consumption validated?

The steps of thematic analysis need to be described in more detail. 

Were the control and intervention groups similar at baseline? 

The qualitative results need a more effective presentation format.

Round 2

Reviewer 2 Report

Comments and Suggestions for Authors

Dear Authors,

Thank you for your corrections. However, the meaning and contribution of the EFAs with low KMOs (0.30-0.45) conducted in both groups at baseline and after follow-up remain unclear, as does their interpretation in the context of the study results. Additionally, the interpretability of the EFA results and the use of colors in the supplementary tables are questionable, and the presentation does not meet data reporting standards.

In my opinion, Exploratory Factor Analysis (EFA) is typically used in two cases: first, to explore the number of factors during a scale validation procedure, and second, to reduce the number of items by grouping them into factors. This process simplifies the data by identifying clusters of items that are highly correlated, thereby capturing the essence of the data in fewer dimensions. EFA can also help in understanding the dimensionality of a construct and in identifying problematic items that don't load well onto any factor, which can inform decisions about item retention or revision.

Please add a detailed explanation of the EFA meaning and interpretation of the results in your study.

Reviewer 4 Report

Comments and Suggestions for Authors

The authors have improved the manuscript considerably. However, I am still concerned about the high percentage of similarity (Percent match: 64%) and the way the results are presented.

Round 3

Reviewer 2 Report

Comments and Suggestions for Authors

Thank you for the corrections.

Author Response

Thank you. In this round of reviewing no additional changes or amendments have been requested from the reviewer. 

Reviewer 4 Report

Comments and Suggestions for Authors

Nothing to declare

Comments on the Quality of English Language

Nothing to declare

Author Response

(The authors gave the same response as above.)
